# In-Season Major Crop-Type Identification for US Cropland from Landsat Images Using Crop-Rotation Pattern and Progressive Data Classification

**Md. Shahinoor Rahman** [1], **Liping Di** [1,*], **Eugene Yu** [1], **Chen Zhang** [1] and **Hossain Mohiuddin** [2]

[1]  Center for Spatial Information Science and Systems (CSISS), George Mason University, Fairfax, VA 22030, USA; mrahma25@gmu.edu (M.S.R.); gyu@gmu.edu (E.Y.); czhang11@gmu.edu (C.Z.)
[2]  School of Urban and Regional Planning, The University of Iowa, Iowa City, IA 52242, USA; hossain.mohiuddin19@gmail.com
[*]  Correspondence: ldi@gmu.edu; Tel.: +1-703-993-6114

**Abstract:** Crop type information at the field level is vital for many types of research and applications. The United States Department of Agriculture (USDA) provides information on crop types for US cropland as a Cropland Data Layer (CDL). However, CDL is only available at the end of the year after the crop growing season. Therefore, CDL is unable to support in-season research and decision-making regarding crop loss estimation, yield estimation, and grain pricing. The USDA mostly relies on field survey and farmers' reports for the ground truth to train image classification models, which is one of the major reasons for the delayed release of CDL. This research aims to use trusted pixels as ground truth to train classification models. Trusted pixels are pixels which follow a specific crop rotation pattern. These trusted pixels are used to train image classification models for the classification of in-season Landsat images to identify major crop types. Six different classification algorithms are investigated and tested to select the best algorithm for this study. The Random Forest algorithm stands out among selected algorithms. This study classified Landsat scenes between May and mid-August for Iowa. The overall agreements of classification results with CDL in 2017 are 84%, 94%, and 96% for May, June, and July, respectively. The classification accuracies have been assessed through 683 ground truth data points collected from the fields. The overall accuracies of single date multi-band image classification are 84%, 89% and 92% for May, June, and July, respectively. The result also shows higher accuracy (94–95%) can be achieved through multi-date image classification compared to single date image classification.

**Keywords:** CDL; CROP DATA LAYER; Landsat; major crop; USA

## 1. Introduction

Crop-specific acreage information, such as crop types with their location, is useful for crop statistics, yield estimation, change in cropland use, cropland environmental dynamics, crop loss assessment, crop pricing, decision support, and policy formulation [1]. Generating yearly or seasonal information for each crop field at the regional scale is not a trivial task in any part of the world. Field survey-based information collection has been replaced using satellite images. Information can be extracted from satellite images multiple times within or outside of the growing season. Therefore, satellite images are useful for crop condition monitoring, growth stage monitoring, and loss assessment. Moreover, the use of satellite images in crop type identification is time and cost effective over vast areas. Worldwide, the National Agricultural Statistics Service (NASS) of the US Department of Agriculture (USDA) probably

made the first attempt to produce unique crop-specific land cover products and dissemination methods annually at a national scale [1,2]. This crop-specific land cover product for the United States (US) is called the Cropland Data Layer (CDL), which has been produced since 1997 for more than 100 unique crop categories across the US; it is delivered annually with 85% to 95% accuracy at a 30–56 m spatial resolution [1]. The usefulness of CDL data has already been proven in practice; for instance, mapping crop types using CDL time series data [3], identifying agricultural production areas [4,5], examining the relationship between agricultural chemical exposure and disease [6], crop specific inundation mapping [7], yield estimation [8,9], crop growth monitoring [10], pure crop pixel identification [11], land use transformation [12–14], agricultural rotational patterns [15,16], chemical/environmental studies for cropland [17,18], crop carbon dynamics [19,20], and cyberinfrastructure for agroecosystem modeling [21]. Therefore, an accurate and timely cropland data layer can facilitate many applications and research in the agriculture sector.

Satellite remote sensing is useful for mapping both crop types and crop rotation patterns. The synoptic and repeated collection of data for vast areas is the main advantage of the application of satellite remote sensing in crop mapping [22–25]. Optical data are one of the most attractive choices for crop mapping among satellite remote sensing options because they offer vegetation indices, frequent revisit, adequate spatiotemporal resolutions, and options for free of charge data distribution [26]. Waldhoff et al. [24] mapped the crop rotation pattern at a catchment scale in Germany for eight consecutive years (2008–2015). At first, they identified crop types through supervised classification of images from multiple sensors (e.g., Landsat, ASTER, SPOT) for each year. They extracted crop rotation patterns at the field level for eight consecutive years from crop maps. Martínez-Casasnovas et al. [22] mapped multi-year cropping patterns between 1993 and 2000 using Landsat images in Spain. Although their approach is not suitable to extract specific sequence of cropping patterns at the field level over the years, spatial distribution of cropping patterns in study area can be visualized. The combination of remote sensing-derived landcover and census data has also been utilized to map the cropping pattern at a half-degree grid in China [23]. Panigrahy and Sharma utilized a similar approach of Waldhoff et al. [24] to map three years of crop rotation patterns in west Bengal of India using the Indian Remote Sensing Satellite (IRS) LISS-I [27]. Conrad et al. [28] mapped crop types and crop rotation using classification and regression trees (CARTs) in central Asia. They utilized MODIS optical bands and derived products such as NDVI. Sonobe et al. [26] mapped crop cover with very high accuracy using multi-temporal Landsat 8 images and derived products in Hokkaido, Japan. Landsat images have been one of the most attractive options due to its high spatial and temporal resolution, which is very suitable for crop mapping [25,26,29,30]. Many derived vegetation indices such as NDVI, EVI, LAI, and SAVI have been utilized along with original Landsat bands to ensure better accuracy in crop classification [25,29–31]. A wide range of classification algorithms such as support vector machine (SVM), random forest (RF), maximum likelihood, spectral angle mapper (SAM) has been utilized for crop mapping [25,26,28,30]. Most of the crop mapping approaches around the world utilize ground truth collected from the field to train the classification models.

Similar to the approaches for crop mapping using satellite remote sensing images, the CDL program in the US also uses moderate spatial resolution satellite images, NASS June Agriculture Survey data and other ancillary data such as National Landcover Data [2]. NASS mainly uses in-season Landsat images to identify major crop types across the US through supervised image classification. Supervised learning requires a training dataset to train image classification models. Thus, the NASS CDL program used ground truth data collected during the June Agricultural Survey (JAS) to train the model for supervised image classification from 1997–2005 [2]. The JAS is a field-based annual survey program of NASS which collects data for 11,000 individual one square-mile sample segments using an area sampling frame. Information on crop types and acreage of all agricultural activities occurring on the primary sampling units (PSU) is collected during field visits [2,32]. In this survey program, approximately 85,000 agricultural and non-agricultural land use tracts are identified within the sampled segments in a given year [33]. The manual digitization requirement of the sampled field

boundary is one of the major drawbacks of JAS segment data [2]. Thus, this field-based ground sample collection for training data is time consuming, costly and tedious. Therefore, the Common Land Unit (CLU) data from the Farm Service Agency (FSA) replaced the JAS segment data in 2017 [2]. The CLU data are a standardized GIS data layer generated from farmers' reports on crop types and acreage sent to 2300 FSA county offices [34,35]. Since this new approach depends on farmers' reports and construction of GIS layer from reported crop types, it may be time and cost intensive. Therefore, the current study proposed a new solution for the training dataset construction based on crop rotation patterns instead of field surveys and depending on the farmers' reports. Usually, CDL is released after a few months of the growing season, approximately at the end of each year [1]. Therefore, the current CDL may not be able to support many research activities which require early season crop information, such as crop-specific inundation mapping, rapid yield estimation, crop specific loss estimation, grain pricing, crop-specific irrigation need, grain supply, and flood policy. Early season crop type identification will facilitate many types of research and applications in the agriculture sector.

From the above discussion, it is evident that CDL is unique and very useful data on US croplands, which contributes to many research activities and applications in this field. However, there are two major drawbacks: the time and cost-extensive training sample collection and the delayed release of the data at the end of the year; these drawbacks hinder the use of CDL in in-season research. Therefore, this study specifically aims to address these two weaknesses of CDL and propose a solution to overcome them. This study aims to extract major crop rotation patterns from historic CDL instead of mapping crop rotation patterns directly from satellite images. The goal of this study is to create in-season CDL based on crop rotation patterns and Landsat image classification with similar accuracy of current CDL.

## 2. Materials and Methods

### 2.1. Study Area

The study area, Iowa State, is a midwestern state in the United States (US) which is located between Missouri and Mississippi rivers (Figure 1). Geographically, Iowa is part of the Great Plains, and about 90% of the lands of this state are devoted to agriculture because of the high fertility of the soils [36]. Iowa ranks second in the nation for agriculture production. This state is also second in total agricultural exports with farmer's exporting more than $10 billion worth of agricultural products in 2013. The state is also considered as the corn belt region of the US due to its high proportion of corn cultivation. According to the Iowa agricultural statistics 2017, corn, soybean, and alfalfa are the top three agricultural products by value of sale $8.5 billion, $5.2 billion, and $0.75 billion, respectively [36], and hence an ideal study area for this research. Every year the usual planting of major crops begins in the middle of April and harvesting starts from the second week of September in Iowa [37].

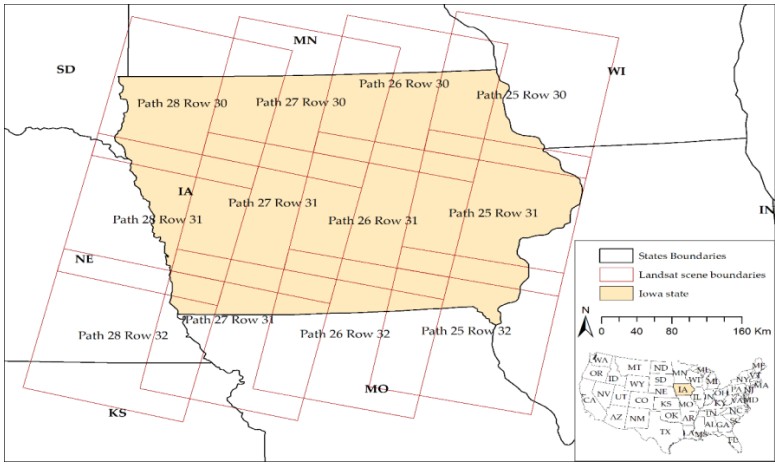

**Figure 1.** Landsat scene boundaries superimposed on the study area.

Figure 2 illustrates the distribution of landcover categories and percentage share of crop types in 2017. Around two-thirds of the land area are used as crop field in Iowa (Figure 2a). Forest, grass, developed areas, and waterbodies cover 9%, 14%, 7%, and 1% of the state land, respectively. Land use distribution of cropland in 2017 shows corn and soybean were dominant crops in 2017. These two crops together cover 96% of cropland (Figure 2b). Alfalfa was planted only on two percent of the cropland. Other crops such as oats, wheat, rye, millet, potatoes, vegetables were together planted only on 2% of the cropland. Therefore, this study focuses only on three major crops: corn, soybean, and alfalfa considering the significant presence of these crops in the study area. Other crops are ignored for two reasons: firstly, it is very difficult to find trusted pixels for these crop types for the study area; secondly, these crop types cannot be considered as major crops in Iowa because of their negligible individual share of the crop production of the state.

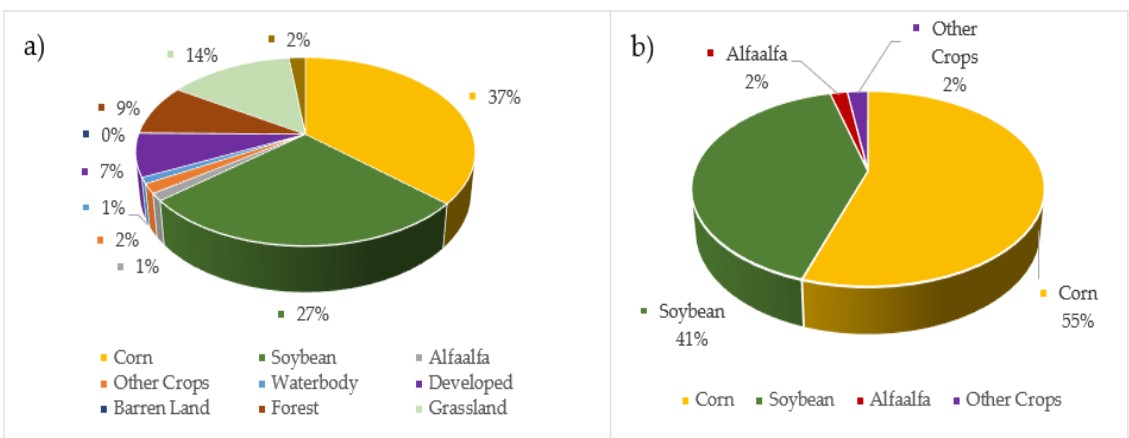

**Figure 2.** (**a**) Distribution of Land cover in Iowa in 2017; (**b**) Major crop share of Iowa in 2017.

## 2.2. Data

Two datasets, Cropland Data Layer (CDL) and Landsat 8 Operational Land Imager (OLI), are used for this study. Trusted pixels for each targeted landcover type are extracted from CDL time series data from 2007 to 2017 downloaded from CropScape (https://nassgeodata.gmu.edu/CropScape/) for the study area. CropScape is a web service of the US Department of Agriculture (USDA) for US geospatial cropland data products [38]. Landsat scenes are downloaded from the USGS Earth Explorer. A total of 12 Landsat scenes, path 25 to path 28 and row 30 to row 32, are required to cover all of Iowa (Figure 1). Due to the growing season of major crops in this state, Landsat scenes between late April and mid-August are downloaded. Six Landsat bands, from band-2 to band-7, are selected for each scene considering the usability of these bands for landcover classification. In addition, three derived products, Normalized Difference Vegetation Index (NDVI), Normalized Difference Water Index (NDWI), and Normalized Difference Built-up Index (NDBI), are added to the image stack. Therefore, an image stack of nine layers, including six Landsat original bands and three derived products, is used as input for the classification. The reason behind the use of three derived products is to reduce the misclassification among landcover types to ensure a better classification result.

## 2.3. Trusted Pixel Identification

Trusted pixels are pixels which maintain a specific crop rotation pattern. In this study, trusted pixels refer to these pixels for which crop types are known from the past use of these pixels. These pixels can be used as substitutes for ground truth data, either from field survey or farmer reports of crop plantation. This trusted pixel approach reduced the cost and time associated with field-based ground truth collections. Figure 3 illustrates two major patterns of crop rotation from 2007 to 2016. An alternate crop rotation pattern indicates that the same crop is planted in a given field in an alternate year. Figure 3a shows soybean and corn alternative patterns, which indicates if a field had corn last year,

then it has soybean this year. Similarly, a monoculture pattern can illustrate a pattern where a field always has the same crop in the past ten years. The field with red marked in Figure 3b shows a monoculture pattern of corn from 2007 to 2016. Pixels which follow these two patterns are considered trusted pixels. Moreover, most of the non-crop landcover types do not change frequently. Therefore, trusted pixels can be found for non-crop land covers through this pattern. These trusted pixels are useful to know the current year crop types at their location.

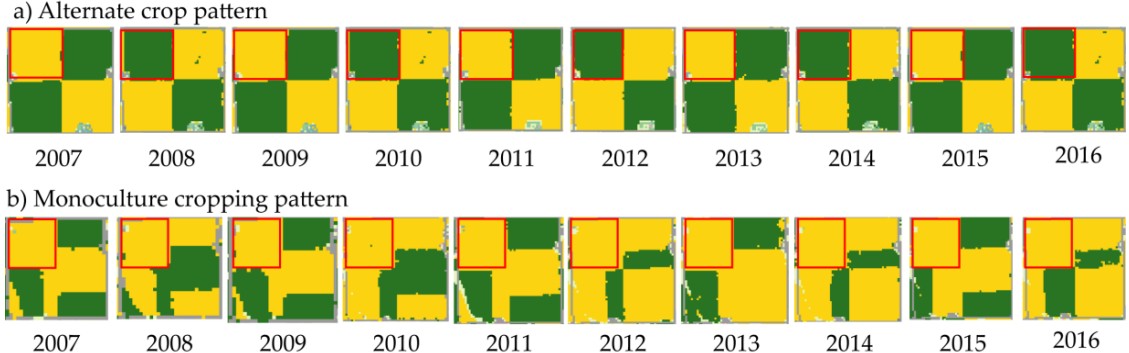

**Figure 3.** Major crop rotation patterns in Iowa state; (**a**) Alternate crop rotation pattern; (**b**) Monoculture cropping pattern.

Trusted pixels are extracted for 2017 using historical CDL from 2007 to 2016 based on two rotation patterns over ten years. Figure 4 illustrates trusted pixels of different land cover types (Figure 4a) and CDL in 2017 (Figure 4b) for the same extent in Iowa. These trusted pixels are used as ground truth to train image classification model for 2017 in-season Landsat imageries. It is important to test the accuracy of these trusted pixels before using them as ground truth. In total, 1023 sample pixels from trusted pixels are selected across the state using stratified random sampling for accuracy assessment. The confusion matrix in Table 1 shows the agreement between these trusted pixels and 2017 CDL data. The overall agreement of nine landcover types is 97%. Kappa measures also show significant agreement (94.5%) of these trusted pixels. Since croplands are the main focus of this research, the user accuracy and the producer accuracy of trusted pixels are 97% and 94%, respectively, for corn. Likewise, soybean has 94% for both user and producer accuracy. Only alfalfa has moderate accuracy where producer accuracy is reported as 78%, and user accuracy is 70%. It can be noted that a crop type which is not very common in an area may lead to lower accuracy of trusted pixels compared to a more dominant crop type. Since the percentage of alfalfa crop covers only 2% of arable land, the accuracy of alfalfa may have minimal impact on overall classification. Because of the high accuracy, trusted pixels can be utilized as ground truth to train the classification model for in-season crop identification. Since long-term CDL is available for US croplands, this study utilized historic CDL data to extract trusted pixels based on major crop rotation pattern. Although CDL data may not be available in many parts of the world, crop rotation patterns can be identified through multi-year crop mapping using remote sensing data [24].

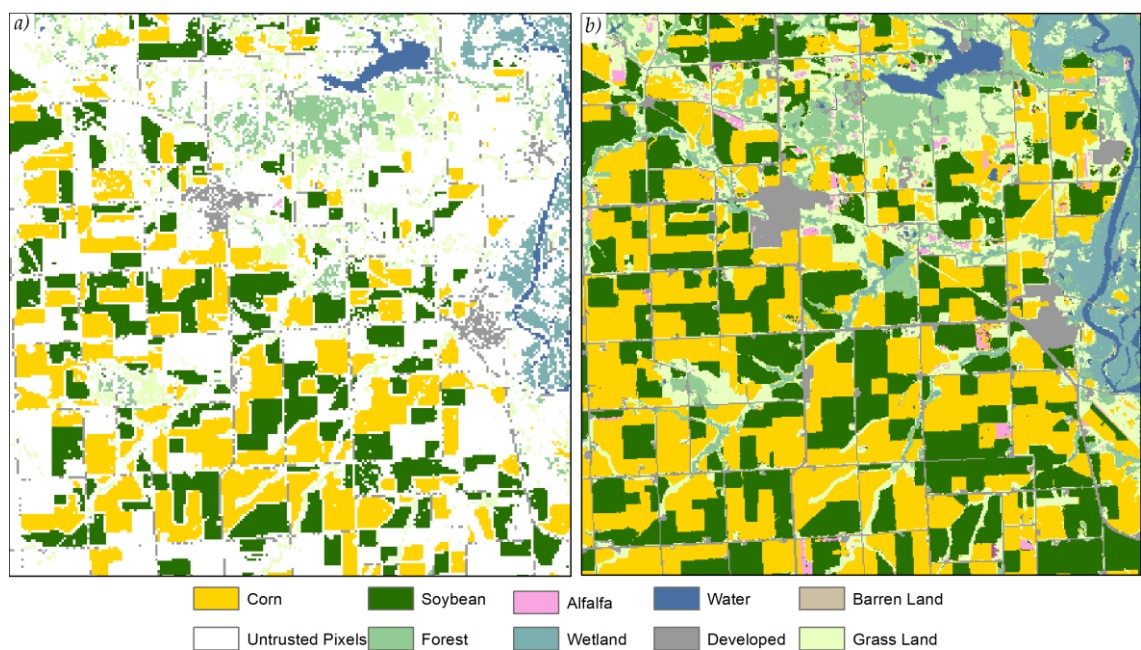

**Figure 4.** (**a**) Land cover types of trusted pixels in 2017 (**b**) Cropland Data Layer (CDL) of 2017.

**Table 1.** The accuracy of trusted pixels in Iowa.

| Class | Corn | Soybean | Alfalfa | Water | Developed | Barren | Forest | Grass Land | Wetland | Total | User Accuracy |
|---|---|---|---|---|---|---|---|---|---|---|---|
| Corn | 334 | 9 | 1 | 0 | 0 | 0 | 0 | 0 | 0 | 344 | 0.97 |
| Soybean | 17 | 270 | 1 | 0 | 0 | 0 | 0 | 0 | 0 | 288 | 0.94 |
| Alfalfa | 2 | 0 | 7 | 0 | 0 | 0 | 0 | 0 | 0 | 9 | 0.70 |
| Water | 0 | 0 | 0 | 15 | 0 | 0 | 0 | 0 | 0 | 15 | 1.0 |
| Developed | 0 | 0 | 0 | 0 | 104 | 0 | 0 | 0 | 0 | 104 | 1.0 |
| Barren | 0 | 0 | 0 | 0 | 0 | 10 | 0 | 0 | 0 | 10 | 1.0 |
| Forest | 0 | 0 | 0 | 0 | 0 | 0 | 139 | 0 | 1 | 140 | 0.99 |
| Grass Land | 1 | 0 | 0 | 0 | 0 | 0 | 1 | 101 | 0 | 103 | 0.98 |
| Wetland | 0 | 0 | 0 | 0 | 0 | 0 | 0 | 0 | 10 | 10 | 1.0 |
| Total | 354 | 279 | 9 | 15 | 104 | 10 | 140 | 101 | 11 | 1023 | |
| Producer Accuracy | 0.94 | 0.94 | 0.78 | 1.0 | 1.0 | 1.0 | 0.99 | 1.0 | 0.91 | | 0.97 |
| Kappa | | | | | | | | | | | 0.945 |

## 2.4. Choice of Classifiers

There are many algorithms available in supervised learning for image classification. However, there is no single algorithm which is better than others for all problems. Wolpert and Macready stated in the "No Free Lunch" theorem that there is no such thing as the "best" algorithm in machine learning, which means an algorithm may work well for one data set whereas another does not work well [39]. Therefore, for each problem, selection of an appropriate algorithm is important. There are many ways to select the appropriate algorithm for a specific problem. This study aims to evaluate the performance of the algorithm through empirical judgment. Six algorithms are selected from six groups of supervised learning algorithms to test the performance for cropland image classification. Six algorithms—Random Forest [40], Support Vector Classification (SVC) [41], K-Nearest Neighbor (KNN) [42], Gradient Boost (Gboost) [43], Multi-layer Perceptron (MLP) [44], and Gaussian Naive Bayes (GNB) [45]—are selected from ensemble methods (forests of randomized trees), support vector machine, nearest neighbor, boosting algorithm, neural network models, and probability-based Naïve Bayes, respectively. All images of study areas from 2007 are classified separately using these algorithms. Therefore, single date multi-band and multi-date multi-band data are investigated for the classification of cropland. Single date multi-band is a stack of Landsat bands on a specific day. Multi-band multi-date is an image stack of Landsat bands from multiple dates. The spatial agreements between classified

images with CDL of 2017 are assessed. The study choses best classifiers for this study based on the performance of the classifiers

### 2.5. Post-Classification Process

In general, it is evident that image classification results have a salt-and-pepper effect. This is mainly because of mixed pixels, which have multiple land covers within a pixel. Therefore, the signature of these pixels differs from the signature of pure pixels for a land cover class. Figure 5a shows the result of the classified image where some isolated pixels appear as a different class than the class of homogenous patch. The assumption here is a cropland unit contains only the same crop. Crop land units are extracted based on the land use homogeneity of the last five years CDL. Thus, salt-and-pepper like isolated pixels are removed from a cropland unit using the majority vote. Crop type is assigned to all pixels within a cropland unit based on the class of the majority the pixels of that cropland unit. Figure 5b illustrates the improvement for the result of crop types of cropland units from the classification result in Figure 5a. This post-classification processing result improves the crop type identification at the field level.

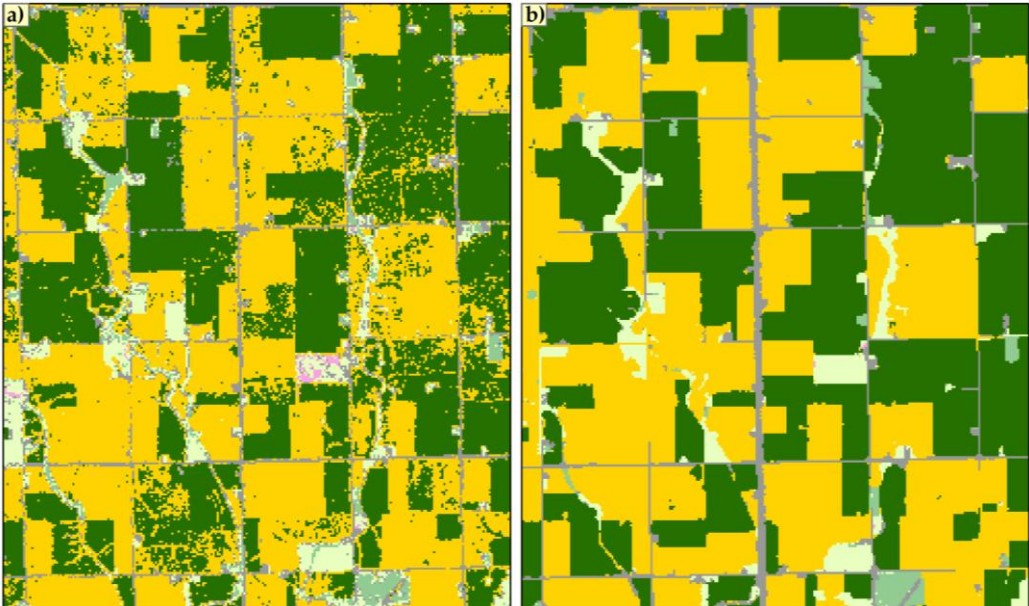

**Figure 5.** (**a**) Classified image; (**b**) Improved result after post-classification processing.

### 2.6. Ground Truth Collection for Validation

Ground truth is important to validate classification accuracy since this study focuses on major crop type identification through supervised classification of remote sensed images. Ground truths are collected on crop fields through field surveys for the validation of in-season crop type identification. A purposeful sampling process is undertaken for this study for several reasons. Firstly, the study area is vast; therefore, it is difficult to follow any random sampling in the whole state. Secondly, the objective is to collect samples only over cropland. Thirdly, purposeful sampling is convenient with limited resource and manpower. Ground truth data points are collected by taking geotagged cellphone (iphone 7s device) photographs along major roads. Highways are selected from Cerro Gordo County in the north, Pottawattamie County in the west. Interstate 80, Interstate 380, Highway 20, and Highway 169 cover the central part of Iowa. Samples of 683 crop fields are collected from more than 600 miles of major roads. Figure 6 shows the spatial distribution of collected samples and crop types for these selected fields.

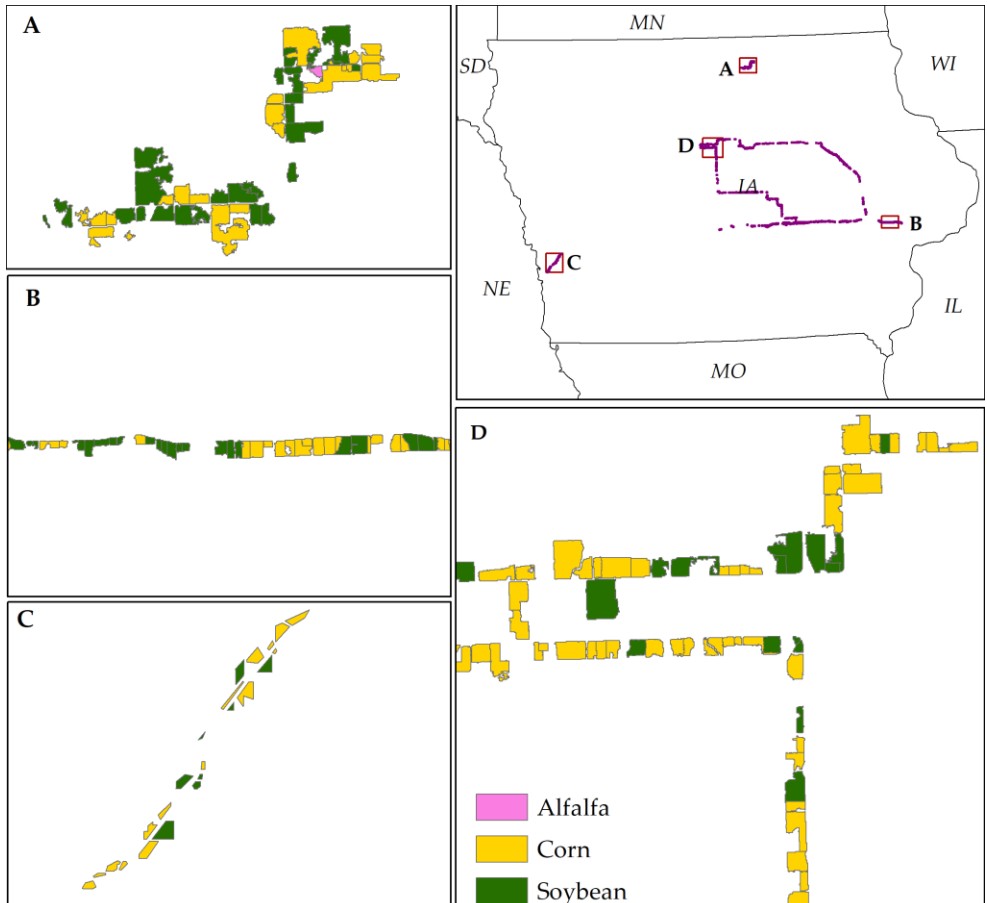

**Figure 6.** Spatial distribution of ground truth for the classification result validation. Panels( **A**–**D**) are the blowup of parts of sampling fields.

## 3. Results

### 3.1. Performance of the Selected Classifiers

Figure 7 illustrates the agreement of selected algorithms for all 12 scenes. Three algorithms—Random forest, SVC, and Gboost—show better agreement than the other three algorithms in most of the scenes. However, agreement varies scene to scene even for the same algorithm. For instance, agreement of the three algorithms for path 26 row 30 in May is more than 85%, whereas the agreemet of path 26 row 32 is just above 70%. Similarly, agreements of all selected algorithms are below 90% in July for the scene Path 25 Row 32, but more than 90% for Path 25 Row 31. Another important aspect is that the agreements increased from May to July for all algorithms in most of the cases. Therefore, crops are more easily differentiated in their mature stage (in July) than in the early stage (in May). The performances of these three algorithms—Random Forest, SVC, and Gboost—are also steady in the sense that they have better accuracy as growing season progresses. However, the performance of the other algorithms fluctuates and does not follow a specific trend. Therefore, Random forest, SVC, and Gboost are more reliable than others. Since June data have better separability than May data, and July data have better separability than June data, the progressive data classification may provide a better result. Progressive data refer to adding more data in the feature space for image classification.

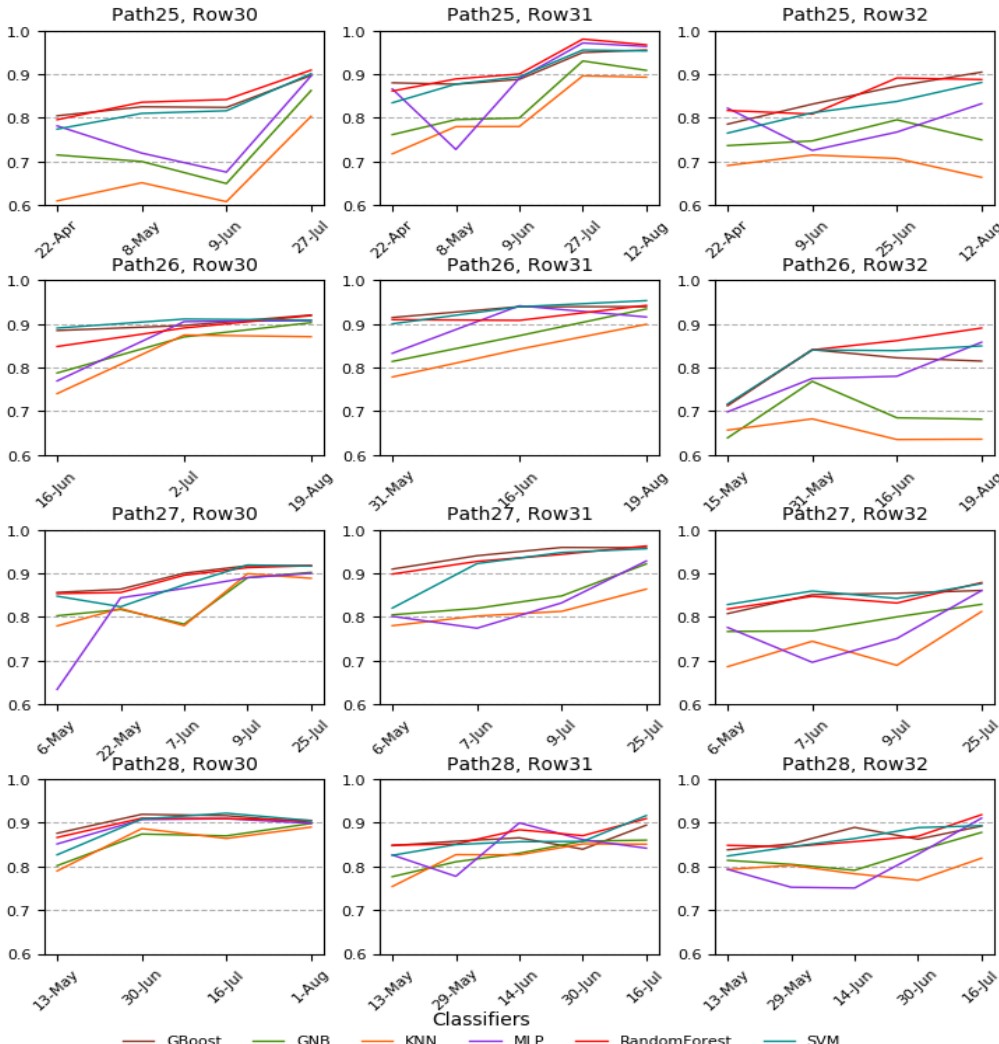

**Figure 7.** Classification accuracy of different classifiers in different Landsat scenes.

Figure 8 shows descriptive statistics for the performance of selected classification algorithms for Landsat scenes. The overall performances of KNN, MLP, and GNB are low compared to the other three classifiers. This result is also supported by the findings of the other researchers who utilized different algorithms for crop mapping from remote sensing images [24,25,30,46]. The Random Forest and SVM have better performance compared to other classifiers in crop mapping [24,30]. For instance, Ouzemou et al. reported best overall accuracy (89.26%) among all classifiers utilized for crop mapping in Morocco [30]. The SVM shows the highest mean performance compared to other classifiers in two scenes (Path 26 Row 30 and Path 27 Row 32). However, it has lower performance than the Random Forest and Gradient boost classifiers in all other scenes. Among 12 scenes, both the Random Forest and the Gradient Boost classifiers have their mean highest performance in five scenes. Therefore, the performances of the Random Forest and the Gradient Boost classifiers are almost similar. The mean overall accuracies of the Random Forest, the Gradient Boost, and the SVM are 88%, 88%, and 87%, respectively, for the 12 scenes. The median accuracies of the Random Forest, the Gradient Boost, and the SVM are 89%, 88%, and 87%, respectively. Although these three classifiers have almost similar performance, this study chooses the Random Forest algorithm for early season crop types identification considering overall highest mean and median accuracy across all 12 scenes. Moreover, the Random Forest algorithm required less computation time than SVM (i.e., minutes compared to hours).

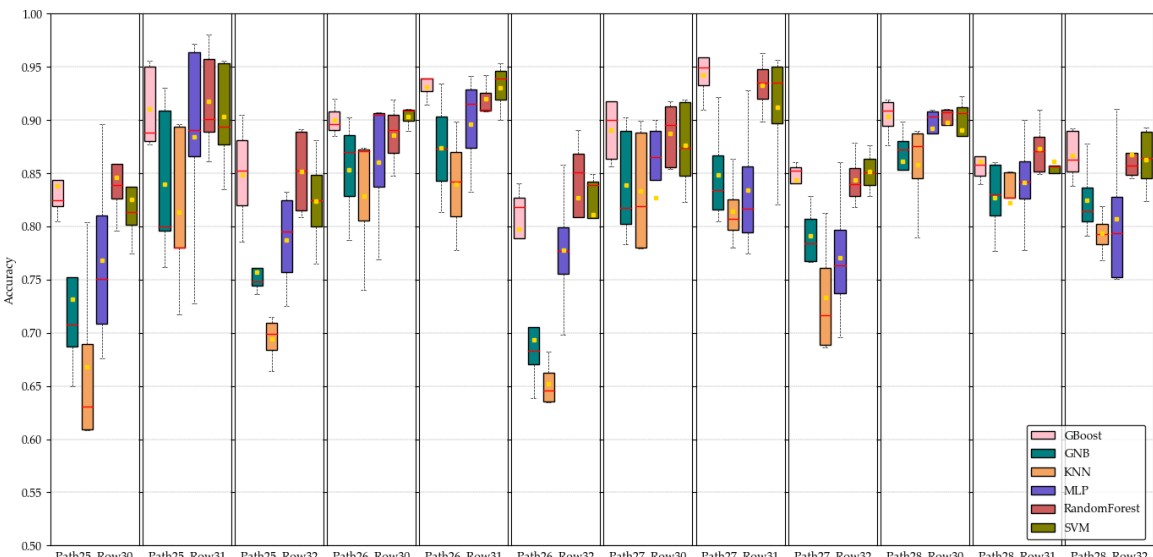

**Figure 8.** Box plots of the performance of selected classification algorithms for Landsat scenes. The upper and lower bounds of the box indicate 75 and 25 percentiles of the distribution, respectively. The mean (square mark), median (bar at the middle of the box), minimum–maximum bound (dash line) of each Landsat scene are also shown in the figure.

### 3.2. Classification Result of Single Date Multi-Bands

Landsat images of all 12 scenes between May and July are classified and processed for major crop type identification. Monthly mosaics are created from classified images in a month to cover all of Iowa. In total, 10,000 sample pixels are selected over cropland using stratified random sampling to assess the agreement between classified landcover and USDA CDL for 2017. Since only three major crops are identified for the study area, samples are taken only on these three croplands. This study aims to examine the agreement between CDL and classification results. Figures A1–A3 illustrate the comparison between classification results and CDL 2017 for May, June, and July, respectively. Four blowups from four locations a, b, c, and d are shown in each figure for the visual comparison of classification result (a1–d1) with CDL (a2–d2). The classification results are improved from early season to late season. The overall agreements between the classification results and CDL are 84%, 91%, and 95% in May, June, and Jul, respectively (Tables A1–A3). Similarly, the kappa statistic is another measure for accuracy assessment, which evaluates the classification values compared to values assigned by chance. Kappa values are in the range of 0 to 1; 0 indicates no agreement between the classified image and the reference image, and 1 indicates both the classified and the reference image are identical. The kappa statistics of the agreements are 0.70, 0.81, and 0.90 in May, June, and July, respectively, indicating the gradual improvement of the result (Tables A1–A3). In May, corn has both higher user accuracy (0.87) and producer accuracy (0.88) than soybean's user accuracy (0.82) and producer accuracy (0.87) (Table A1). The user accuracy and producer accuracy of alfalfa are only 50% and 60%, respectively (Table A2). Corn has higher user accuracy but lower producer accuracy than soybean in June. Both user and producer accuracies of alfalfa increased to near 70% in the classification of June image. Both user accuracy and producer accuracy of alfalfa are lower compared to corn and soybean, which may be the result of the low accuracy of trusted pixels for alfalfa. Since the success of the major crop type identification is highly dependent on trusted pixel accuracy, this approach may not be effective in an area where trusted pixels have low accuracy. From June to July, there is no significant improvement in producer accuracy for corn and alfalfa, but soybean has higher producer accuracy in July than in June. User accuracy of all three major crops increased significantly from June to July. Corn, soybean, and alfalfa have producer accuracies of 98%, 98%, and 77% respectively, in July (Table A3). Therefore, it is evident that overall agreement with CDL 2017 increases as crops grow.

### 3.3. Classification Result of Multi-Date Multi-Bands 2017 Images

Similar to single date multi-band image classification, this research also investigated the classification outcome using multi-date image stacks. An image stack of multi-date and multi-band is the combination of Landsat bands of multiple dates for classification. An image stack of multiple dates increases the dimension of data for classification algorithms. This study considered multiple combinations of images from different dates available within the growing season. For instance, images from May combine with images from June when data in June are available. Figure 9 compares the classification result of multi-dates with CDL 2017. From the visual interpretation of Figure 9, the classification result from the combination of May and July images is closer to CDL than the combination of May and June images. Likewise, the classification result from the combination of May, June, and July images is almost identical to CDL. The agreement between classification result and CDL has been evaluated through 10,000 randomly sampled points. Tables 2 and 3 show agreement between classification result and CDL through confusion matrix. Table 2 indicates the overall agreement of the classification result from the combination of May and June images with CDL is 94%, and the kappa value is 0.87. This multi-date image classification result has a higher agreement (94%) compared to the overall agreement of single date June images classification result. Likewise, multi-date image classification has an overall better agreement, 96% in July, compared to the result of single date July image classification result of 95%. The kappa value is also improved in multidate image classification from single date image classification. Therefore, higher data dimension from multi-date imageries can improve the classification accuracy in general for major crop identification.

**Table 2.** Agreement between multi-date classification result of May–June images and CDL in 2017.

| Class Name | C | S | A | OC | W | D | B | F | G | WL | Total | UA |
|:---:|:---:|:---:|:---:|:---:|:---:|:---:|:---:|:---:|:---:|:---:|:---:|:---:|
| C | 5289 | 276 | 3 | 4 | 0 | 5 | 0 | 6 | 13 | 5 | 5601 | 0.94 |
| S | 300 | 4047 | 4 | 5 | 1 | 3 | 0 | 4 | 10 | 2 | 4376 | 0.92 |
| A | 2 | 0 | 16 | 4 | 0 | 0 | 0 | 0 | 1 | 0 | 23 | 0.70 |
| Total | 5591 | 4323 | 23 | 13 | 1 | 8 | 0 | 10 | 24 | 7 | 10,000 | |
| PA | 0.95 | 0.94 | 0.70 | | | | | Overall Agreement 0.94 | | | Kappa Statistics 0.87 | |

C = Corn; S = Soybean; A = Alfalfa; OC = Other Crop; W = Water; D = Developed; B = Baran Land; F = Forest; G = Grass Land; WL = Wetland; UA = User Agreement; PA = Producer Agreement.

**Table 3.** Agreement between multi-date classification result of May–June–July images and CDL in 2017.

| Class Name | C | S | A | OC | W | D | B | F | G | WL | Total | UA |
|:---:|:---:|:---:|:---:|:---:|:---:|:---:|:---:|:---:|:---:|:---:|:---:|:---:|
| C | 5390 | 139 | 5 | 1 | 0 | 11 | 0 | 5 | 21 | 5 | 5577 | 0.97 |
| S | 216 | 4142 | 3 | 4 | 1 | 9 | 0 | 5 | 15 | 3 | 4398 | 0.94 |
| A | 0 | 0 | 19 | 1 | 0 | 0 | 0 | 0 | 5 | 0 | 25 | 0.76 |
| Total | 5606 | 4281 | 27 | 6 | 1 | 20 | 0 | 10 | 41 | 8 | 10,000 | |
| PA | 0.96 | 0.97 | 0.70 | | | | | Overall Agreement 0.96 | | | Kappa Statistics 0.91 | |

C = Corn; S = Soybean; A = Alfalfa; OC = Other Crop; W = Water; D = Developed; B = Baran Land; F = Forest; G = Grass Land; WL = Wetland; UA = User Agreement; PA = Producer Agreement.

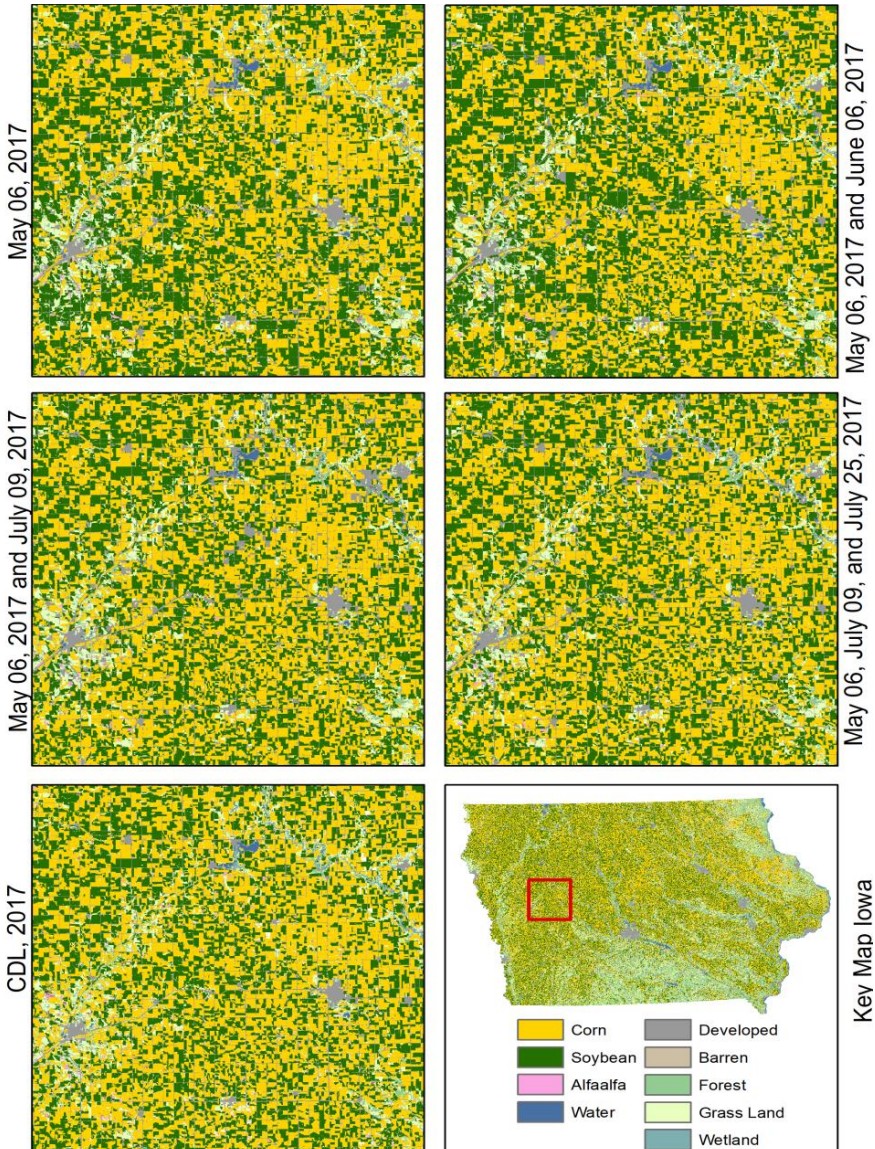

**Figure 9.** Comparison of the classification results of multi-bands multi-date images with CDL for 2017.

### 3.4. Current Year In-Season Classification Result

The goal of this study is to identify in-season major crop types. The previous section compares the in-season crop identification results with USDA CDL. Therefore, the result is the agreement between the classification outcome and CDL, which is not the actual classification accuracy. Section 3.4 discusses the current year (2018) classification results through the validation with ground truth collected from fields. Both single date multi-band and multi-date multi-band image classification results are discussed here. Figure 10 shows major crop types identified through single date multi-band image classification in three different months of growing season. Four blowups are also shown from four different locations in each month. Table 4 shows classification accuracy increases over time; the overall accuracy is 84%, 80%, and 92%, respectively, in May, June, and July. The kappa values, 0.67 in May, 0.76 in June, and 0.83 in July, indicate the improvement in major crop type identification. The classification result is improved because of the higher separability among crop types in the late growing season compared to the early growing season. The improvement in user accuracy and producer accuracy also indicates a decrease in misclassification among crop types (Table 5). The user accuracy of corn increases from 83% in May to 93% in July. Likewise, producer accuracy increases from 94% in May to 97% in July. Although both

producer and user accuracy for soybean are lower compared to corn, a significant improvement can be seen in both user accuracy and producer accuracy from May through July. Based on the agreement between classification results and CDL 2017 (Tables A1–A3 and Tables 2 and 3), both the user accuracy and the producer accuracy of alfalfa are lower compared to those of corn and soybean in 2018 as well. Ouzemou et al. [30] also reported similar problem in crop mapping in Morocco, where they found very low classification accuracies for alfalfa. Asgarian et al. [25] also reported that the classification accuracy is lower for alfalfa than other crops in Iran because of the misclassification of vegetation and other crop types. Therefore, the performance of the crop type identification is consistently low for crops which are not very common in the study area. Although other states in the US may have different major crop types, the similar crop rotation patterns can be found in most of the states. Therefore, the approach of major crop identification utilized in this study may be applicable to other states. Since crop plantation is a diverse and complex phenomenon widely affected by regional factors, rotation pattern identification methods can benefit from the use of a more complete historical data and more sophisticated algorithms. Since the approach of this heavily depends on trusted pixels, this approach may not be effective for these areas where crop plantation does not follow a pattern.

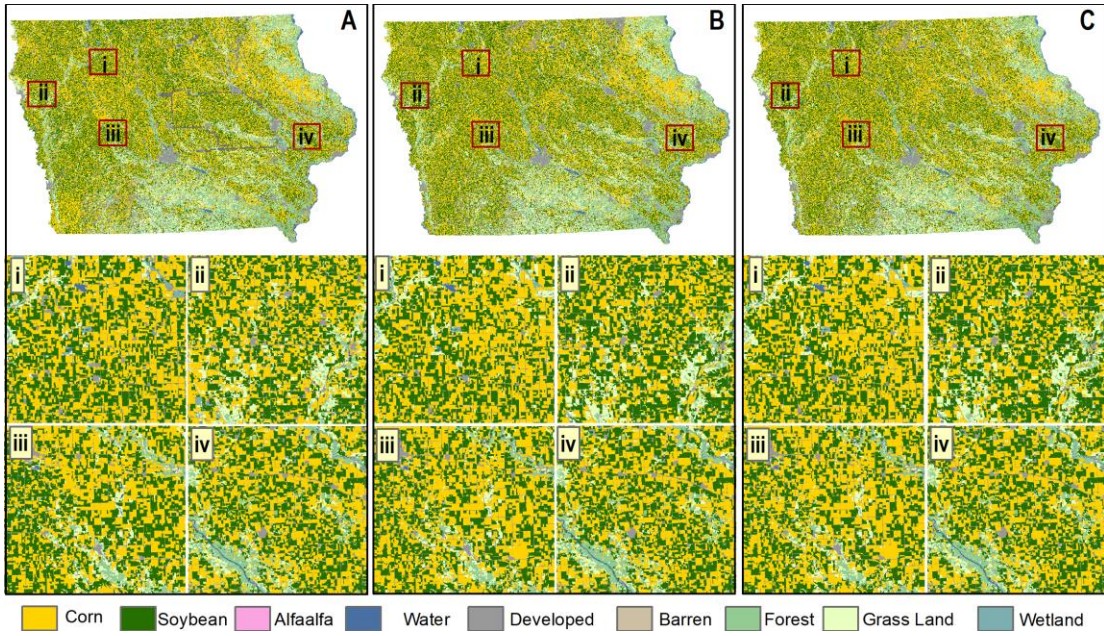

**Figure 10.** Identification of major crops through single date multi-band image classifications; panel (**A**–**C**) show the result for May, June, and July, respectively.

Like the single date multi-band classification, this study also investigated the multi-date multi-band classification approach. Figure 11 illustrates the result of in-season major crop types identification through image classification of multi-date multiband images. Panels A, B, C, and D show enlarged detail of four different locations in Iowa. Boxes 1, 2, and 3 in each panel show the classification result from May, May–June multi-date, and May–June–July multi-date images, respectively. From the visual interpretation of Figure 11, multi-date image classification shows higher accuracies compared to single date image classification. The overall classification accuracy is 94% for May–June multidate image classification, which is higher than the single date classification accuracy 89% in June. A higher kappa value of 0.86 for multidate classification compared to a kappa value of 0.83 for single date classification in June indicates better agreement with multi-date image classification. The highest overall accuracy (95%) and kappa value (0.88) are achieved from the classification of May–June–July multi-date images. User accuracies and producer accuracies are also improved in multi-date image classification. Both producer accuracy and user accuracy for corn and soybean are above 95% and 90%, respectively.

**Table 4.** Overall accuracy and a kappa value of major in-season major crop types identification in 2018.

| Accuracy Metrics | Single Date Multi-Band | | | Multidate-Date Multi-Band | |
|---|---|---|---|---|---|
| | May | June | July | May–June | May–June–July |
| Overall Accuracy | 0.84 | 0.89 | 0.92 | 0.94 | 0.95 |
| Kappa Value | 0.67 | 0.76 | 0.83 | 0.86 | 0.88 |

**Table 5.** User accuracy and producer accuracy of major in-season major crop type identification in 2018.

| Crop Types | Accuracy | Single Date Multi-Band | | | Multidate-Date Multi-Band | |
|---|---|---|---|---|---|---|
| | | May | June | July | May–June | May–June–July |
| Corn | User Accuracy | 0.84 | 0.90 | 0.93 | 0.95 | 0.96 |
| | Producer Accuracy | 0.94 | 0.96 | 0.97 | 0.98 | 0.98 |
| Soybean | User Accuracy | 0.85 | 0.88 | 0.91 | 0.92 | 0.93 |
| | Producer Accuracy | 0.75 | 0.82 | 0.87 | 0.89 | 0.91 |
| Alfalfa | User Accuracy | 0.57 | 0.64 | 0.79 | 0.79 | 0.86 |
| | Producer Accuracy | 1.00 | 1.00 | 1.00 | 0.92 | 1.00 |

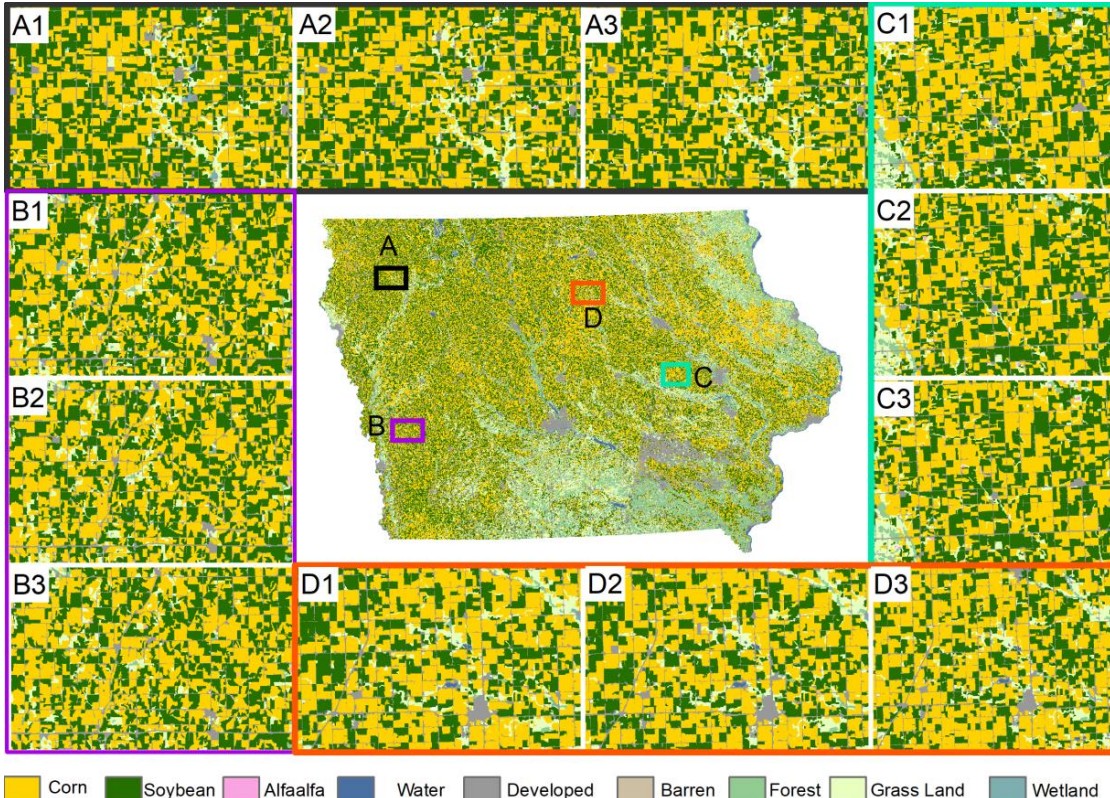

**Figure 11.** Identification of major crops through multi-date multi-band image classifications. (**A1**): 16 May 2018; (**A2**): 16 May, and 1 June 2018; (**A3**): 16 May, 1 June, and 3 July 2018; (**B1**): 16 May 2018; (**B2**): 16 May, and 1 June 2018; (**B3**): 16 May 1 June, and 3 July 2018; (**C1**): 18 May 2018; (**C2**): 18 May and 3 June 2018; (**C3**): 18 May, 3 June, and 22 Augus 2018. (**D1**): 18 May 2018; (**D2**): 18 May, and 3 June 2018; (**D3**): 18 May, 3 June, and 22 August 2018.

Table 6 shows the summary of the results of in-season major crop type identification through satellite image classification. This study processed in-season Landsat images for the years 2017 and 2018 for Iowa. The results of the 2017 image classification are assessed through the spatial agreement with 2017 CDL. The actual accuracy of the results of current year in-season major crop identification are validated through ground truth collected from the field. Both overall agreement and the kappa value between classification results and 2017 CDL are increased from May through July. Similarly, the in-season classification accuracies are improved from 0.84 in May to 0.92 in July. The improvement in classification accuracies indicates that crop types are easily identifiable as the season progress. Multi-date multi-band image classification shows a better result both in June and July compared to single-date multi-band image classification. Therefore, progressive data classification can be more helpful for accurate crop type identification.

**Table 6.** Summary table of the results of in-season major crop type identification.

| Month | Single-Date Multi-Band | | | | | | Multidate-Date Multi-Band | | | |
|---|---|---|---|---|---|---|---|---|---|---|
| | May | | June | | July | | June | | July | |
| | Overall | Kappa | Overall | Kappa | Overall | Kappa | Overall | Kappa | Overall | Kappa |
| Agreement with CDL 2017 | 0.84 | 0.7 | 0.91 | 0.81 | 0.95 | 0.9 | 0.94 | 0.87 | 0.96 | 0.91 |
| Classification Accuracy in 2018 | 0.84 | 0.67 | 0.89 | 0.76 | 0.92 | 0.83 | 0.94 | 0.86 | 0.95 | 0.88 |

## 4. Conclusions

In-season crop type identification is crucial for research and decision-making processes. In-season crop type identification supports many in-season studies such as crop loss estimation, yield prediction, agriculture trade, and environmental research. The USDA provides crop-specific information each year. However, crop-specific information is only available at the end of the year, many months after the crop growing season. One of the major reasons for this delay is because of the collection of ground truth to train the image classification model. This study utilized trusted pixels identified from major crop rotation patterns as training data to train the image classification model. The overall accuracy of the trusted pixel is 97%, which is very high considering there are nine major land cover classes including crops. This study selected the Random Forest algorithm as a classifier due to the better performance over pre-selected six classifiers. In-season images from both 2017 and 2018 are classified in this study. The classification results from 2017 are compared with CDL to investigate overall agreement. The results show very high overall agreement up to 96% with most of the cases above 90%. The CDL program reports classification accuracy between 85% and 95%. The agreement with CDL is not the actual classification accuracy. In-season classification results of 2018 are validated through the ground truth collected by field survey. The results show classification accuracy between 84% and 95%, which is similar to CDL. Multi-date image classification shows higher accuracy compared to single date image classification. The overall accuracies of multi-date image classification are 94% and 95% in June and July, respectively. The only limitation of in-season crop identification is cloud contamination in optical imageries. Although this study focuses on Landsat data, optical data from sentinel mission may also be useful for major crop identification. The Sentinel mission has more frequent revisit capability compared to Landsat which increases the chances to obtain more in-season cloud-free images. This approach to major crop-type identification can be useful in areas that have a certain/fixed patterns of crop rotation. Since different areas tend to have different patterns of crop rotation, the patterns adopted and utilized in this study may not be representative of the patterns in other areas. Therefore, it is necessary to investigate the major crop rotation patterns before the extraction of trusted pixels. It can be concluded that major crop type identification through the proposed method can achieve a similar accuracy of CDL. In-season crop research can benefit through the proposed methodology of in-season crop classification.

**Author Contributions:** M.S.R. and L.D. conceptualized and designed the experiment; M.S.R. performed the analysis. H.M., E.Y., and C.Z. collected ground truth, processed geotagged photographs, and prepared reference map of ground truth for the validation of 2018 image classification. E.Y., and C.Z. contributed in the design of data processing framework. M.S.R. wrote the whole article. L.D. supervised the overall research.

**Funding:** This study was supported by a grant from the U.S. National Science Foundation (Grant # CNS-1739705 PI: L.D.).

**Acknowledgments:** We would like to thank the United States Geological Survey (USGS) for their Landsat archive in the Earth Explorer portal. We are grateful to the National Agricultural Statistics Service (NASS) of the United States Department of Agriculture (USDA) for historical CDL data. We would like to express our special thanks to Zhiqi Yu of CSISS, GMU, for his comments and sugestions in initial data preprocessing. We are greatful to Zhi Chen and Reuben Grandon of the University of Iowa for their assistance during field data collection in Iowa. We are also thankful to anonymous reviewers for their comments and sugestions for the improvement of the manuscript.

**Conflicts of Interest:** The authors declare no conflict of interest.

## Appendix A

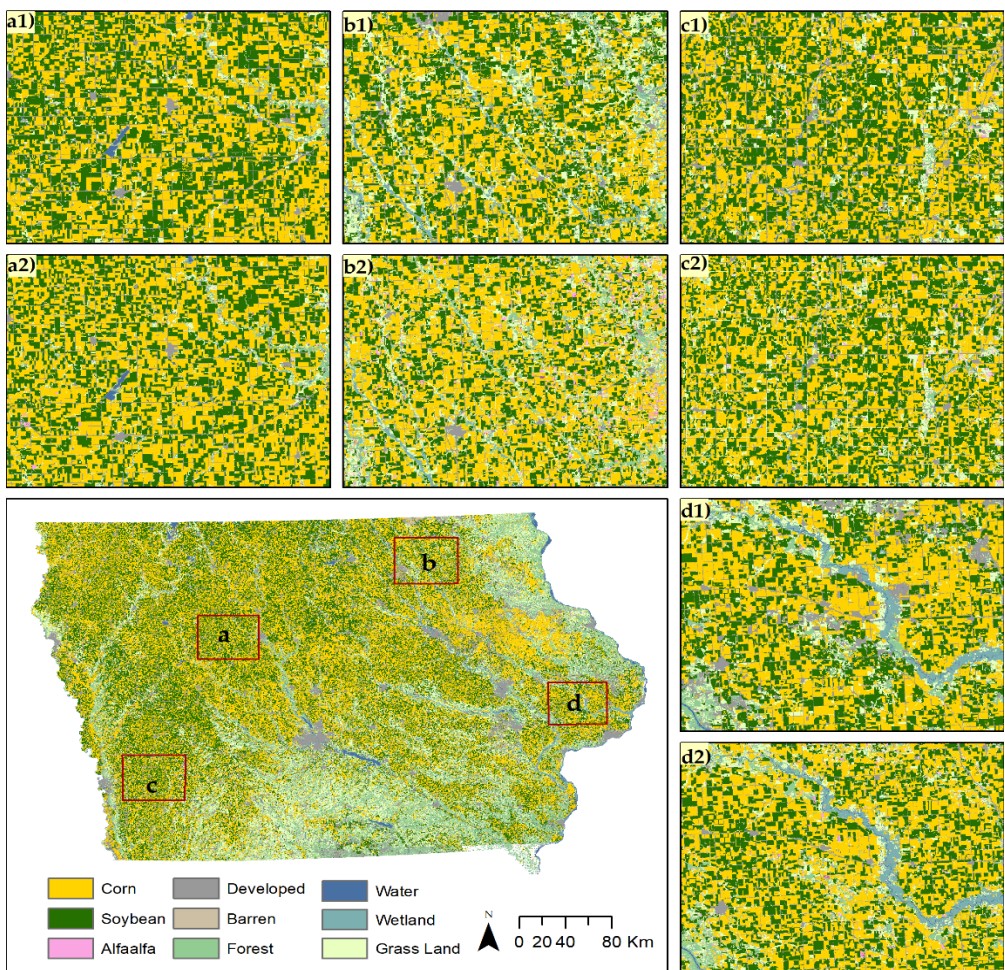

**Figure A1.** Classification results of May and CDL in 2017; (**a1**–**d1**) are snapshots from the classification result, and (**a2**–**d2**) are snapshots from CDL at locations a, b, c, and d respectively.

**Table A1.** Agreement between classification result of May and CDL in 2017.

| Class Name | C | S | A | OC | W | D | B | F | G | WL | Total | UA |
|---|---|---|---|---|---|---|---|---|---|---|---|---|
| C | 4813 | 528 | 7 | 10 | 3 | 75 | 1 | 15 | 95 | 12 | 5559 | 0.87 |
| S | 635 | 3599 | 9 | 8 | 1 | 53 | 1 | 9 | 73 | 5 | 4393 | 0.82 |
| A | 4 | 5 | 24 | 7 | 0 | 1 | 0 | 2 | 5 | 0 | 48 | 0.50 |
| Total | 5452 | 4132 | 40 | 25 | 4 | 129 | 2 | 26 | 173 | 17 | 10,000 | |
| PA | 0.88 | 0.87 | 0.60 | | | | | Overall Agreement 0.84 | | | Kappa Statistic 0.70 | |

C = Corn; S = Soybean; A = Alfalfa; OC = Other Crop; W = Water; D = Developed; B = Baran Land; F = Forest; G = Grass Land; WL = Wetland; UA = User Agreement; PA = Producer Agreement.

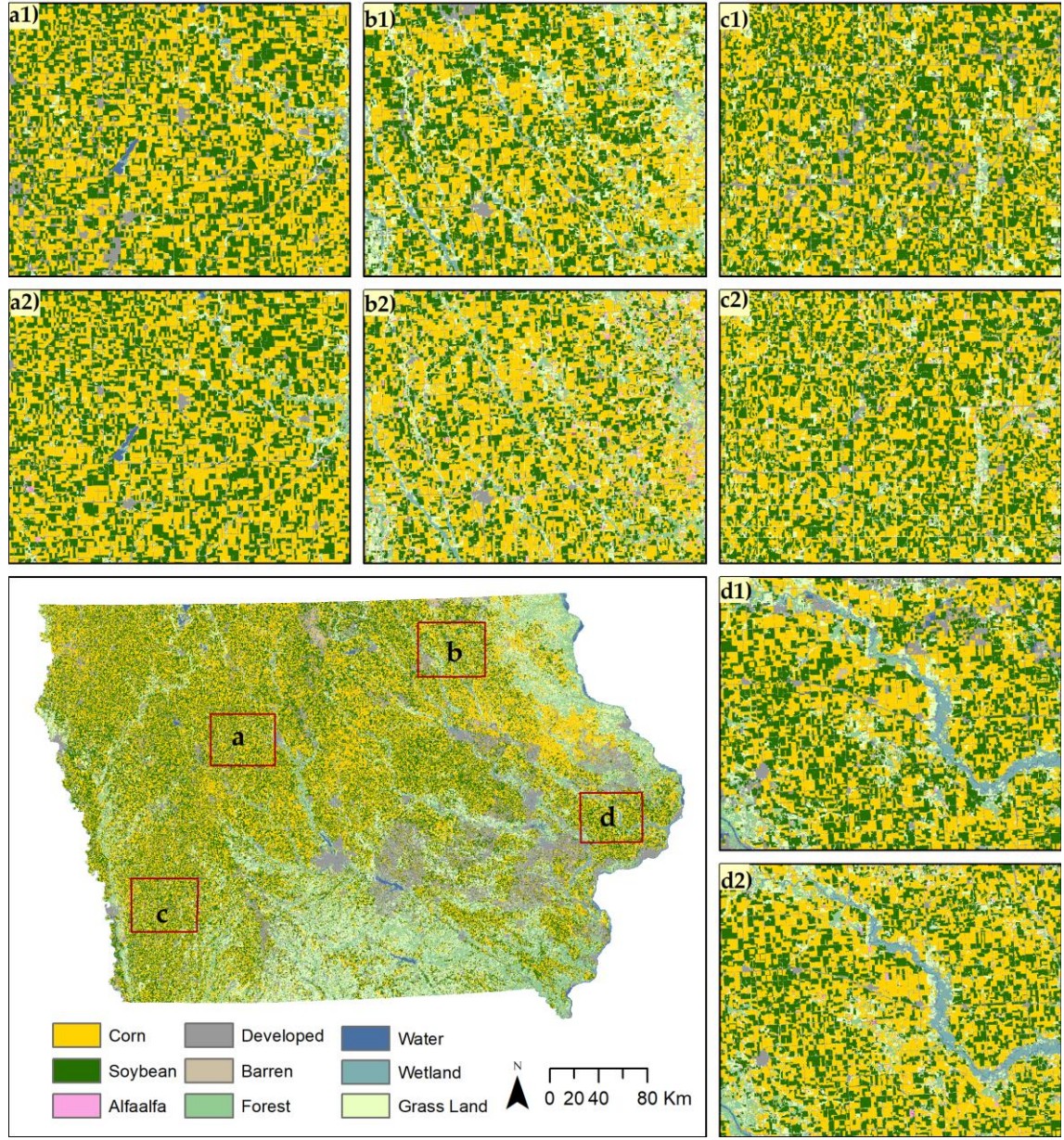

**Figure A2.** The classification results for June and CDL in 2017; (**a1**–**d1**) are snapshots from classification result, and (**a2**–**d2**) are snapshots from CDL at locations a, b, c, and d, respectively.

**Table A2.** Agreement between classification results of June and CDL in 2017.

| Class Name | C | S | A | OC | W | D | B | F | G | WL | Total | UA |
|---|---|---|---|---|---|---|---|---|---|---|---|---|
| C | 5071 | 265 | 4 | 3 | 0 | 18 | 2 | 27 | 55 | 15 | 5460 | 0.93 |
| S | 467 | 3966 | 8 | 6 | 2 | 8 | 3 | 5 | 38 | 1 | 4504 | 0.88 |
| A | 1 | 1 | 25 | 5 | 0 | 0 | 1 | 0 | 3 | 0 | 36 | 0.69 |
| Total | 5539 | 4232 | 37 | 14 | 2 | 26 | 6 | 32 | 96 | 16 | 10,000 | |
| PA | 0.92 | 0.94 | 0.68 | | | | | Overall Agreement 0.91 | | | Kappa Statistic 0.81 | |

C = Corn; S = Soybean; A = Alfalfa; OC = Other Crop; W = Water; D = Developed; B = Baran Land; F = Forest; G = Grass Land; WL = Wetland; UA = User Agreement; PA = Producer Agreement.

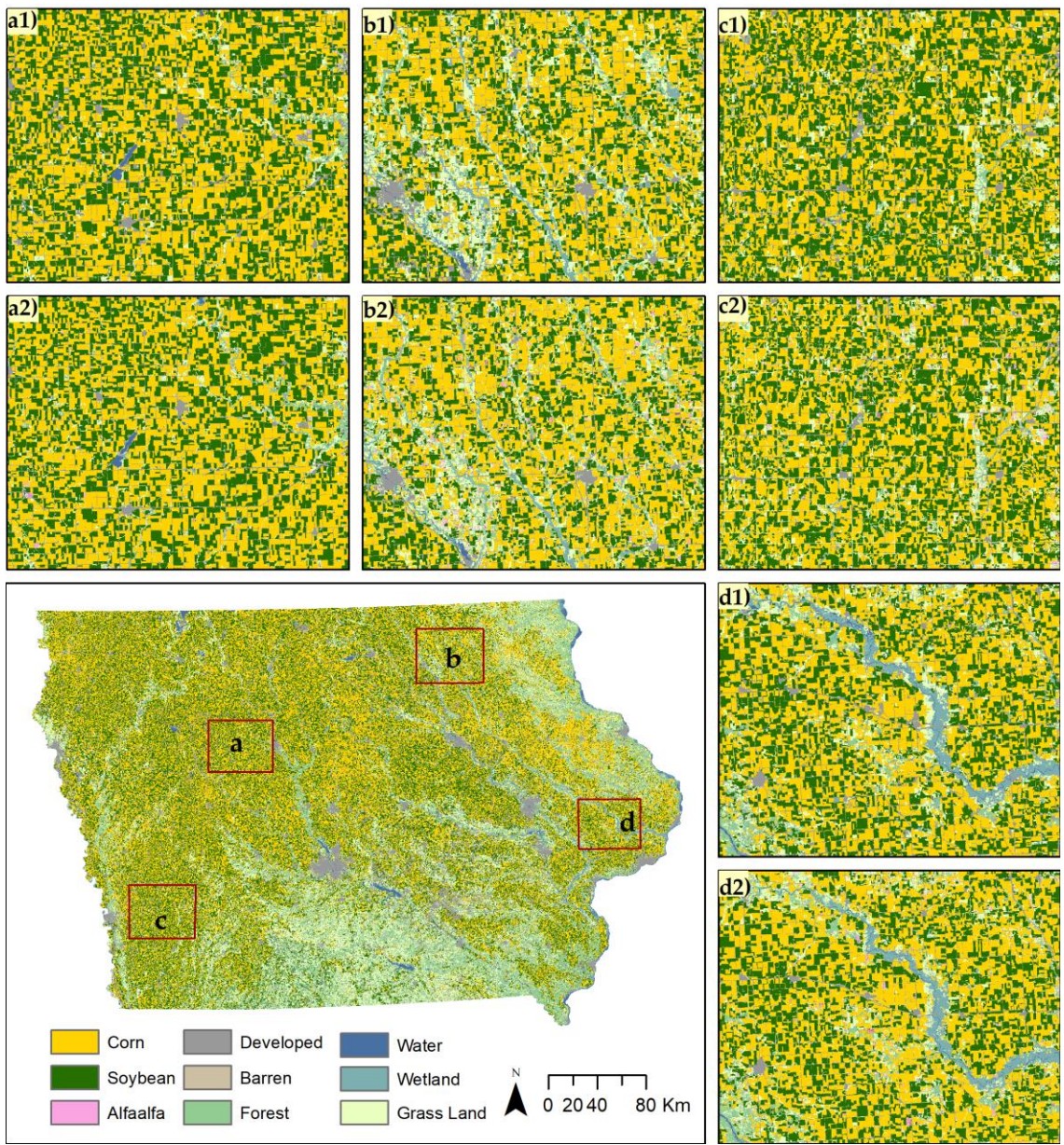

**Figure A3.** Classification result for July and CDL in 2017; (**a1–d1**) are snapshots from classification result, and (**a2–d2**) are snapshots from CDL at locations a, b, c, and d, respectively.

**Table A3.** Agreement between classification result of July and CDL in 2017.

| Class Name | C | S | A | OC | W | D | B | F | G | WL | Total | UA |
|---|---|---|---|---|---|---|---|---|---|---|---|---|
| C | 5302 | 98 | 5 | 2 | 3 | 32 | 0 | 45 | 40 | 26 | 5553 | 0.95 |
| S | 110 | 4150 | 3 | 18 | 2 | 37 | 0 | 9 | 77 | 2 | 4408 | 0.94 |
| A | 1 | 2 | 27 | 2 | 0 | 2 | 1 | 1 | 3 | 0 | 39 | 0.69 |
| Total | 5413 | 4250 | 35 | 22 | 5 | 71 | 1 | 55 | 120 | 28 | 10,000 | |
| PA | 0.98 | 0.98 | 0.77 | | | | | Overall Agreement 0.95 | | | Kappa Statistics 0.90 | |

C = Corn; S = Soybean; A = Alfalfa; OC = Other Crop; W = Water; D = Developed; B = Baran Land; F = Forest; G = Grass Land; WL = Wetland; UA = User Agreement; PA = Producer Agreement.

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
