# Peer review of "In-Season Major Crop-Type Identification for US Cropland from Landsat Images Using Crop-Rotation Pattern and Progressive Data Classification"

_agriculture, doi:10.3390/agriculture9010017_

Round 1
Reviewer 1 Report
In this study the concept of using trusted pixels identified from major crop rotation patterns as training data for satellite image classification is applied as a more cost and time efficient method to provide information on crop types for large areas (e.g. US cropland) before the end of the cultivation period. The proposed methodology is applied in an extensive and very detailed case study in Iowa State.
The paper is very interesting and very well written. The methodology is scientifically sound and very well described and justified. The results are also reasonable and the conclusions are supported by the results.
My only important comment is related with a possible limitation of the proposed methodology. Specifically, the proposed concept seems to be very efficient and suitable for the case study region, which is dominated (almost entirely covered) be three major crops, corn, soybean, and alfalfa that follow specific crop rotation patterns. However, this is a very specific case. It is not certain if the same methodology would be efficient / suitable for other regions with more diverse and complex crop patterns. For example, from my experience from south European countries, I doubt that it would be possible to identify adequate numbers of trusted pixels. This fact does not limit the quality and the importance of this study; though, I believe that the possible limitations should be discussed and it should be also explained what are the next steps / future study on this matter.
Apart from the above important comment I only have some more mostly minor comments. For convenience I included them in the pdf file of the manuscript.
Based on the above, my suggestion is that this manuscript should make minor revisions.

Author Response
Dear Reviewer
Thank you for the constructive comments. We followed the suggestions and responded to each comment below. Our responses are in bold font. We believe this revised version of the manuscript is improved from the previous version.
In this study the concept of using trusted pixels identified from major crop rotation patterns as training data for satellite image classification is applied as a more cost and time efficient method to provide information on crop types for large areas (e.g. US cropland) before the end of the cultivation period. The proposed methodology is applied in an extensive and very detailed case study in Iowa State.
The paper is very interesting and very well written. The methodology is scientifically sound and very well described and justified. The results are also reasonable and the conclusions are supported by the results.
Thank you so much for these encouraging words.
My only important comment is related with a possible limitation of the proposed methodology. Specifically, the proposed concept seems to be very efficient and suitable for the case study region, which is dominated (almost entirely covered) be three major crops, corn, soybean, and alfalfa that follow specific crop rotation patterns. However, this is a very specific case. It is not certain if the same methodology would be efficient / suitable for other regions with more diverse and complex crop patterns. For example, from my experience from south European countries, I doubt that it would be possible to identify adequate numbers of trusted pixels. This fact does not limit the quality and the importance of this study; though, I believe that the possible limitations should be discussed and it should be also explained what are the next steps / future study on this matter.
Thank you so much for pointing an important issue. Actually, this study completely based on trusted pixels extracted through crop rotation patterns. Crop rotation patterns and major crop types may not be same in other locations. Therefore, major crop rotation patterns are needed to be investigated for the desired case before the extraction of the trusted pixels. This discussion is added to the conclusion in the revised version of this manuscript.
Apart from the above important comment I only have some more mostly minor comments. For convenience I included them in the pdf file of the manuscript.
All grammatical errors mentioned by the reviewer have been taken care of in the revised version of the manuscript.
1. This is mostly results and I believe that should be presented in the results section (figure 5 and 6 also).
These two paragraphs are moved to the result section as the reviewer suggested.
2. A short presentation of the various statistics / Accuracy Metrics used would be helpful.
Table 9 and related discussions are added at the end of the result section as summary discussion.

Reviewer 2 Report
Generally sound paper some suggestions to improve English.
Line 20 stands out better than outstands
Line 41 better to add is called the
Line 56 ground truth "information or data" before collected
Line 87 Remove the in Iowa's the
Line 88 Remove richest and productive suggest "because of the high fertility of the soils"
Line 90 farmers' not farmer's exporting better than export
Line 92 remove vast arable land "high proportion of corn cultivation"
Line 93 the top three by value:
Line 95 begins not begin
Line 116 the whole of
Line 117 replace considering with due to
Line 123 products better than product
Line 128 suggest these pixels can be used as substitutes for ground truth data
Line 181 differentiated much better than spreadable
Line 212 salt and pepper not "piper"
Line 216 salt, pepper not piper
Line 218 the majority of the pixels
Line 219 improvement to the result
Line 220 improves the crop
Line 225 remote sensed images better than sensing
Line 226 field surveys better than survey
line 227 replace considered with undertaken
Line 231 from a phone device (should describe make and model)
Line 245 the whole of Iowa State. A total of 10,000 (add comma)
Line 252 improve better than are improved
Line 268 remove are between crops and grow
Line 287 stacks better than stack
Line 291 in June is better than Junes are
line 295 remove of before CDL
Line 364 remove "to"
Line 365 rotation patterns better than pattern
Line 379 more frequent
Author Response
Dear Editor
Thank you very much for your encouraging comments. Your detail suggestions are very helpful for the revised version of the manuscript. All suggested grammatical corrections have been taken care of and highlighted in the version of the manuscript.